# Diametaphyseal Distal Forearm Fractures in Children: A STROBE Compliant Comparison of Outcomes of Different Stabilization Techniques Regarding Complications

**DOI:** 10.3390/children10020374

**Published:** 2023-02-14

**Authors:** Andreas D. P. Wollkopf, Florian S. Halbeisen, Stefan G. Holland-Cunz, Johannes Mayr

**Affiliations:** 1Department of Pediatric Surgery, University Children’s Hospital Basel, 4056 Basel, Switzerland; 2Surgical Outcome Research Center Basel, University Hospital Basel, University Basel, 4053 Basel, Switzerland

**Keywords:** diametaphyseal fracture, distal radius, pediatric trauma, ESIN, K-wire

## Abstract

Diametaphyseal forearm fractures are difficult to treat because standard methods for long-bone fracture stabilization in the metaphyseal or diaphyseal regions are less effective in this transition zone. We hypothesized that there is no difference in outcomes between conservative and surgical treatment of diametaphyseal forearm fractures. This retrospective analysis included 132 patients who had undergone treatment for diametaphyseal forearm fracture between 2013 and 2020 at our institution. The primary analysis compared complications occurring in patients treated conservatively with those occurring in patients managed surgically (ESIN, K-wire fixation, KESIN stabilization, or open reduction and plate osteosynthesis). In a subgroup analysis, we compared the two most frequently applied surgical stabilization techniques in distal forearm fractures (i.e., ESIN and K-wire) with conservative treatment. The mean age of the patients at the time of intervention was 9.43 ± 3.78 years (mean ± SD). Most patients were male (91; 68.9%), and 70 of 132 (53.1%) patients underwent surgical stabilization. The rate of re-intervention or complications was similar after conservative and surgical treatment, and ESIN or K-wire fixation achieved comparable complication rates. Recurrent displacement of fragments was the most frequent reason for re-interventions (13 of 15 patients; 86.6%). There was no permanent damage as a result of a complication. The median time of exposure to image intensifier radiation was comparable between ESIN (95.5 s) and K-wire fixation (85.0 s), but significantly lower during conservative treatment (15.0 s; *p* = *0*.001).

## 1. Introduction

The forearm continues to be the most common fracture site in children in Switzerland [1], the rest of Europe, and the world [2,3,4]. Most patients with a distal forearm fracture are treated by closed reduction (CR) and cast application [5]. This approach is a common and less invasive treatment for all types of distal radius fractures [6,7], possibly resulting in fewer re-interventions when compared to surgical management [8]. Conservative treatment tends to provide similar outcomes even without prior reduction, especially in younger children because of their higher remodeling potential [9,10,11]. However, this has to be confirmed in larger trials. On the other hand, CR and casting might prove inferior in obese patients [12]; depending on fracture location and characteristics, selected cases might require surgical management.

The most frequently applied surgical technique for stabilizing diaphyseal forearm fractures is elastic stable intramedullary nailing (ESIN) [13,14,15], while Kirschner (K)-wire fixation is the most commonly applied surgical method to treat metaphyseal fractures of the distal radius [16].

All surgeons at our tertiary-care university hospital aimed to identify potentially unstable diametaphyseal fractures during the operation. They applied the “hand-shaking stability test” after CR of the fracture. If displacement recurred, they opted for surgical stabilization of the fracture. If only minimal displacement of the fragment was evident, a long-arm plaster cast was applied.

Unstable fractures of the diametaphyseal junction of the radius pose a particular challenge [17,18]. First, the steep contact angle of K-wire tips relative to the corticalis hinders the correct placement of two K-wires. These can easily bounce off the corticalis or follow an intramedullary route in the proximal fragment of the radius [18]. Furthermore, in very distal fractures of the forearm, angular stability is difficult to achieve with ESIN. To achieve the required three-point support, the nail should be bent in an S-shaped manner to additionally stabilize the fracture by stretching the intraosseous membrane [19]. This may be challenging if the distance available for the three-point support is short. 

In light of the ongoing debate on treating diametaphyseal forearm fractures as either diaphyseal or metaphyseal fractures [20] or as an entity of their own [21], the optimal treatment method remains to be established. Prior publications focusing on the management of diametaphyseal forearm fractures in children either compared only two methods (Du et al. [22], Lieber et al. [23]), included only a few patients (Kubiak et al. [17], Ebert et al. [24]), or combined metaphyseal and metadiaphyseal fractures (Józsa et al. [20], Sato et al. [21]) or metadiaphyseal and diaphyseal fractures (Kim et al. [25]) in their analyses.

Treating fractures of the diametaphyseal junction of the distal forearm remains difficult in clinical practice, and better knowledge might prove beneficial in terms of avoidable reoperations or complications, undesirable outcomes, or unnecessary exposure to radiation in children. We, therefore, designed a retrospective analysis to establish if there is a superior treatment method for distal diametaphyseal forearm fractures.

The primary aim of this study was to compare conservative and surgical stabilization of diametaphyseal forearm fractures in terms of the risks of unplanned secondary interventions requiring anesthesia. We also analyzed the incidence of other types of postoperative complications. 

The secondary aim of this study was to compare intraoperative and postoperative outcomes associated with either K-wire fixation or stabilization with ESIN. To the best of our knowledge, this study describes the largest patient cohort focusing exclusively on stabilizing distal diametaphyseal forearm fractures in children.

## 2. Materials and Methods

### 2.1. Study Design

We conducted a retrospective STROBE-compliant observational study. The primary outcome was the difference in the number of unplanned and planned reoperations (requiring general anesthesia) and the rate of other complications between conservative and surgical treatment. In the secondary analysis, we compared the differences and characteristics of the two most frequently applied stabilization techniques, i.e., ESIN and K-wire fixation, with conservative management in terms of associated complications.

The Cantonal Ethics Committee approved the study protocol (project ID “2021-00581”). We included children who underwent treatment for a forearm fracture in the distal diametaphyseal junction area (Figure 1) at the UKBB (University Children’s Hospital Basel, Basel, Switzerland) between 1 January 2013, and 31 December 2020.

Patients aged from 2 years to 17 years were eligible for inclusion. For the primary analysis, all patients with a diametaphyseal forearm fracture were registered with their initial treatment at the UKBB. This included 4 cases of revision operations on patients who had initially been treated at other institutions.

Exclusion criteria comprised pathologic and recurrent fractures, fractures of the diaphysis or metaphysis, and epiphyseal separations. We also excluded all patients who withdrew their consent in writing or in any other way.

We only used pre-existing digital data from the UKBB hospital patient management software system and analyzed the following relevant patient data (Table 1):

### 2.2. Measurements

We used the definition of the diametaphysis as proposed by Lieber et al. [23]. The diametaphysis comprises the area of the distal radius located between the squared radio-carpal joint width and the squared diameter of the distal radius in the anterior-posterior (AP) projection (Figure 1). We included all patients whose fracture was located in this diametaphyseal transition zone.

One observer analyzed the radiologic images (ADPW). In difficult decisions, he was supported by a second observer (JM).

To allow the comparison of X-ray images, we defined a “relative fracture location” (RFL). The RFL was calculated by dividing the distance between the fracture site and growth plate of the distal radius by the maximum distal width of the radius (RFLR) or the distal radio-ulnar joint width (RFLDRUJ). The distance of the fracture from the distal radial epiphysis had to be greater than the radial width (RFLR > 1) and smaller than the width of the distal radio-ulnar joint (RFLDRUJ < 1) to qualify as a diametaphyseal fracture (Figure 1). The calculated RFL value was unitless and could thus be used for comparison with other cases. An RFL value in relation to the radial width (RFLR) close to 1 characterized a very distal diametaphyseal fracture.

### 2.3. Additional Analyses

Additional parameters assessed were angular stability throughout the healing process, defined as the difference between the axis deviation measured by X-ray imaging immediately after surgery vs. that measured at the last follow-up examination. We only evaluated deviations in the lateral projection in this study because we regarded them as the clinically most relevant displacements. To investigate whether the fracture location influenced the choice of implants, we checked for an association between the RFL and the type of surgical technique. We also analyzed the total time of exposure to image intensifier radiation per intervention. The main investigator (ADPW) then entered all assessed and calculated data into a Redcap^®^ (Vanderbilt University, Nashville, TN, USA) database for analysis.

### 2.4. Statistical Methods

We analyzed baseline and procedure characteristics descriptively. For normally distributed data, we computed means and standard deviation (SD); otherwise, median and interquartile range (IQR) were reported. To check for differences between treatment groups, we used Fisher’s exact tests for categorical data. For continuous variables, we first used QQ plots and Shapiro–Wilk normality tests to check the normality of the distribution. Since none of the continuous variables tested were normally distributed, we used the Wilcoxon rank-sum test to check for differences between the groups. All statistical analyses were performed using R (R Core Team, http://www.R-project.org/, accessed on 14 February 2023).

## 3. Results

We identified 215 patients who had undergone stabilization of a fracture in the diametaphyseal area of the distal radius or forearm. After accurate measurement of the fracture location, we included 132 patients in this study (Figure 2). Four patients had previously been treated at another hospital and were referred to our hospital for further treatment. In total, 11 board-certified pediatric and orthopedic surgeons and 16 experienced residents performed the operations.

### 3.1. Patient Characteristics

The mean age at the time of treatment was 9.43 ± 3.78 years. Male patients accounted for 91 cases (68.9%). The median length of hospital stay (LOS) was 1 day (IQR: from 1 day to 1 day). In addition, 27 of 132 (20.5%) patients were treated as outpatients, with immediate discharge after CR and casting, followed by a visit to the outpatient fracture clinic the next day.

### 3.2. Fracture Mechanism/Type

Most patients suffered a diametaphyseal forearm injury by either falling on level ground (n = 44) or during sporting events (n = 44) (Table 2). 

We observed no significant difference between affected sides (48.4% right-sided and 51.6% left-sided fractures). Moreover, no bilateral diametaphyseal forearm fractures occurred, but one patient sustained a concomitant fracture of the contralateral distal radius at another location.

Overall, 84 (63.6%) patients exhibited displaced fractures, with the ratio of grossly displaced fractures (defined as ad latus displacement of more than a shaft width of the radius) accounting for 39 of the 84 (46.4%) fractures. Greenstick fractures occurred in 32 of 132 (24.2%) patients.

### 3.3. Type of Stabilization

In total, 47 of 132 (35.6%) fractures were stabilized with K-wire fixation using a metaphyseal K-wire entry point in forty-five children (34.1%) and a transepiphyseal entry point in two (1.5%) patients (Table 3). In comparison, 14 of 132 (10.6%) were treated with ESIN.

### 3.4. Primary Analysis (Surgical vs. Conservative Treatment) 

The rate of re-interventions and incidence of complications were similar in patients treated conservatively and patients undergoing surgery (Table 4). However, patients treated conservatively tended to be more likely to undergo a secondary intervention (7 of 62 patients; 11.3%) than patients treated surgically (4 of 70 patients; 5.7%), but this difference was not statistically significant.

### 3.5. Subgroup Analysis (Conservative vs. ESIN vs. K-Wire Fixation)

We assessed differences between the two most frequently applied surgical stabilization techniques (ESIN and K-wire fixation) and compared them to conservative treatment (CR and cast application). We analyzed the incidence of revision operations (Table 4) and the type and rate of any other complications (Table 5).

A total of 61 patients underwent osteosynthesis by either ESIN or K-wire fixation. We stabilized 14 fractures (23.0%) with ESIN placement and 47 fractures (77.0%) with K-wire fixation. Overall, 62 patients underwent conservative treatment (CR and cast application). Thus, the subgroup analysis contained 123 (61 + 62) of 132 patients (93.2%).

The mean age of these three patient groups (n = 123) was 9.19 ± 3.70 years, which compared well with the mean age of the total study population (n = 132; 9.43 ± 3.78 years).

#### 3.5.1. Complications

We used the Clavien-Dindo classification of surgical complications [27,28] to categorize complications. The primary outcome of this study was defined as any re-intervention performed under general anesthesia, according to a Clavien-Dindo grade IIIb complication. In Table 4, these cases are listed as “re-operation” and are therefore not shown in the complications section. We observed 19 additional complications, of which 16 (84.2%) represented grade I and 3 (15.8%) were grade IIIa complications. None of these complications resulted in permanent damage or restriction of the range of motion (ROM) exceeding 20 degrees at the wrist joint.

The type and severity of complications varied between groups, but the difference was not statistically significant (*p* = 0.721). Of the nineteen patients who suffered an additional complication, six (31.6%) had been treated with CR and cast application, ten (52.6%) with K-wire fixation, and three (15.8%) patients treated with ESIN suffered a complication.

Secondary displacement occurred in 24 of 132 (18.2%) patients and represented the most prevalent complication. We noted nerve lesions in 4 of 132 (3.1%) patients, a tendon injury in one patient, and wound healing disorders in two (1.5%) patients.

Neither of the two patients treated with transepiphyseal K-wire fixation suffered any complications, similar to the four (3.0%) patients treated with K-ESIN.

#### 3.5.2. Secondary Radiologic Parameters/Parameters Potentially Influencing Stability after Reduction of Fracture

The fracture characteristics of patients treated with CR and casting and patients treated with ESIN or K-wire fixation differed significantly. Non-greenstick fractures were more prevalent in the group of patients treated with ESIN or K-wire fixation. All 14 (100%) patients treated with ESIN and 46 of 47 patients (97.9%) treated with K-wire fixation had suffered a fracture with both cortices disrupted, while only 51.6% of patients treated with CR and casting exhibited a non-greenstick fracture. The difference between the conservatively and surgically treated groups was statistically significant (*p* < 0.001). However, there was no statistically significant difference between ESIN and K-wire fixation (*p* = 1.0).

Similarly, the rate of massive displacement (defined as displacement of more than one radial shaft width) and a combined fracture of the distal radius and ulna was significantly lower in patients treated conservatively (*p* = 0.008 and *p* = 0.01, respectively). There was no significant difference in the rate of massive displacement (*p* = 0.73) and incidence of concomitant fracture of the ulna (*p* = 1.0) between children stabilized with ESIN and those managed with K-wire fixation.

#### 3.5.3. Residual Axis Malalignment 

There was a considerable variation of accepted residual axis malalignment after reduction, but no clear tendency related to the age of the child or the surgical technique was evident (Figure 3). 

#### 3.5.4. Total Time of Exposure to Image Intensifier Radiation

The time of exposure to image intensifier radiation associated with ESIN or K-ESIN was slightly but not significantly higher than that associated with K-wire fixation (exposure to ionizing radiation, 95.5 s vs. 85.0 s; *p* = 0.594). The median time of exposure to image intensifier radiation was significantly lower in patients treated conservatively (15.0 s) than in patients treated surgically (*p* = 0.001; Figure 4).

The boxplots show the total time of exposure to image intensifier radiation for the three stabilization techniques. The four patients treated with K-ESIN were combined with the patients undergoing ESIN.

#### 3.5.5. Choice of Treatment Relative to Fracture Location

More distally located diametaphyseal fractures of the radius (values for RFLR close to one) were preferably treated with K-wire fixation. In contrast, fractures located proximally within the diametaphysis were preferably treated with ESIN (*p* < 0.001) (Figure 5).

### 3.6. Revision Interventions

Overall, 15 re-interventions were necessary for 14 patients. Among these fourteen patients, four had initially been treated at another institution, and ten had previously undergone treatment at our hospital. 

The main reason for re-intervention was displacement of fragments (13 of 15 cases, 86.6%), including four patients who had sustained a refracture. One patient was reoperated because of a nerve injury, and another patient underwent reoperation because of an infection at the nail entry site. 

CR and cast immobilization had been the initial treatment in 9 of 15 (60.0%) patients who underwent revision operations. Of the six remaining patients, five had initially been treated with K-wire fixation, and one with ESIN.

Overall, 8 of the 15 patients (53.3%) underwent revision surgery with K-wire stabilization. In total, five patients (33.3%) underwent revision surgery by open reduction and volar plate osteosynthesis of the distal radius. One patient was retreated by CR and cast immobilization, and one by ESIN. In addition, one patient underwent a second revision operation after experiencing a refracture with breakage of the plate 3 months after the first revision operation. Another patient was retreated using volar plate osteosynthesis (see Figure 6 for a case description and X-ray images).

## 4. Discussion

Diametaphyseal forearm fractures are difficult to treat. The optimal treatment modality remains unclear and needs to be defined to guide surgeons in providing the most efficient and safest treatment option for children suffering diametaphyseal forearm fractures. Each of the established methods of stabilizing the diaphysis or metaphysis poses technical difficulties in the diametaphyseal region of the radius, as described in a review by Lieber et al. [18]. Persiani et al. reported that conservative treatment of diametaphyseal forearm fractures results in a lower rate of axis retention [7].

We assessed the rate of re-interventions and complications in children undergoing treatment for a diametaphyseal forearm fracture. The rates of re-intervention after conservative and surgical fracture treatment did not differ significantly. Similarly, the rate of other complications was comparable for the three types of fracture management. We observed a complication rate of 8.3% in terms of necessary re-interventions and 22.7% for overall complications for the whole study population. A subgroup analysis of ESIN and K-wire stabilization compared with CR and cast application did not show any significant differences in the rate of re-intervention and other complications between the groups. These numbers are consistent with reported data on pediatric fractures of the distal radius in general, with re-intervention rates ranging from 5.0% to 6.9% for ESIN, K-wire, and conservative treatment quoted by several authors [13,29,30]. 

In the literature, overall complication rates (including minor complications) are reported to be as high as 33.3% [17,29,31]. The nonsignificantly higher rate of re-interventions in our conservatively treated patients (11.3%) is in accordance with the reports of Persiani et al. and Caruso et al. [7,32]. The main reason for reoperation in our study was secondary displacement in 13 of 15 (86.6%) patients, of whom nine had been treated with CR and cast application beforehand. In their study of 360 children with metaphyseal fractures undergoing cast application after CR, Persiani et al. [7] obtained a re-intervention rate of 14.2% for distal metaphyseal fractures of the radius. Of these 360 children, 21 (5.8%) subsequently underwent revision surgery. The authors identified age as a predisposing factor for revision surgery and hypothesized that the older the child, the less remodeling potential for post-traumatic fracture malalignments remains. Another mechanism might be the regression of soft tissue swelling that occurs in the early postoperative period [33]. However, all authors agree that the quality of the initial reduction is an important [7,33] or even the most important [34] factor in determining the risk of subsequent displacement of fragments.

In their review of the management of diametaphyseal forearm fractures, Kubiak et al. found a higher refracture rate after CR and cast immobilization than after ESIN or K-wire fixation. All refractures occurred in the group managed conservatively, which also had a significantly higher complication rate in general (*p* = 0.012) [17].

Kruppa et al. reported a low complication rate of 8.9% in 201 children undergoing treatment with ESIN for distal forearm fractures [13]. Of these 201 patients, only 6.9% required re-interventions, which is comparable to our value (1 of 14 patients; 7.1%). They described a slightly higher complication rate (12.5%) when focusing on diametaphyseal fractures. Patients treated primarily with cast application accounted for 9 of 15 (60.0%) re-interventions [13].

When comparing the radiologic fracture parameters measured intra-operatively to those measured at the final follow-up, we observed a significant difference in fracture characteristics between patients undergoing CR and casting and those whose fractures were managed surgically. We hypothesize that parameters such as severe displacement and concomitant fracture of the ulna are indicative of a massive deforming force. This might encourage surgeons to choose a more invasive approach. On the other hand, the two most frequently applied surgical techniques, i.e., ESIN and K-wire fixation, turned out to be not statistically significant in terms of the rate of complications. The inevitable selection bias of this retrospective study favors fractures considered more stable by the clinician to be managed by CR and casting. This must be kept in mind.

Accepted axis malalignment among our patients varied considerably, but no clear tendency related to the age of the child or surgical technique emerged. Considering the higher correction potential in younger children [32], one would assume the accepted value for residual axial malalignment to be higher in younger children and in children managed with CR and conservative treatment, but our findings did not support this. Even larger axis deviations did not result in functional restrictions in our study population. This is in line with the findings of Crawford et al. [35], who reported full recovery of ROM in all patients receiving a cast without prior CR. In their study of 51 patients with overriding metaphyseal forearm fractures, cast immobilization alone was sufficient to achieve excellent clinical outcomes, which they attributed to the high remodeling and self-correction potential in children. Newer studies by Marson et al. [8] and Barvelink et al. [36] were similarly able to show that unreduced fractures in younger children heal well, indicating that surgical management and even CR of distal forearm fractures may not be necessary in most young children. The follow-up assessment at our institution was not standardized, however, and ROM restrictions might thus have been underestimated.

The median exposure time to image intensifier radiation was significantly shorter in patients treated with CR and cast application (median time: 15 s) than in those undergoing surgery (median time: 95.5 s for ESIN and 85.0 s for K-wire fixation). Other stabilization techniques (e.g., plate osteosynthesis [32]) may be associated with shorter exposures to ionizing radiation. Moreover, training programs on the safe use of ionizing radiation might help reduce the time of an image intensifier per intervention, regardless of the stabilization technique applied [37].

Diametaphyseal fractures of the radius located more distally tended to be stabilized with K-wire fixation, whereas fractures within the proximal diametaphysis of the distal radius were preferably treated with ESIN. This observation has also been described by Józsa et al. [20] in their study of 196 children undergoing surgical treatment for diametaphyseal forearm fractures with either K-wire fixation, dorsally inserted titanium elastic stable intramedullary nails (DESIN), or short titanium elastic stable intramedullary nails (SESIN). The authors showed a significant correlation between the use of K-wire fixation in fractures closer to the radiocarpal joint and the use of modified ESIN techniques in more proximally located fractures. Ebert et al. [24] observed the same trend in their assessment of diametaphyseal forearm fractures in 88 patients suffering from proximal diametaphyseal fractures treated predominantly with intramedullary stabilization techniques and distal diametaphyseal fractures treated with K-wires. They described a significantly higher rate of poor outcomes (defined as an axis deviation of more than 10° at follow-up after 4 weeks) when K-wires were used in proximal fractures and ESIN in distal fractures. The authors proposed to develop an algorithm based on a subdivision of the diametaphysis into proximal, central, and distal diametaphyses.

Thus, surgeons should select the most appropriate method of stabilization based on the location of the diametaphyseal fracture of the radius in relation to the radiocarpal joint. Subdividing the diametaphysis into a distal and proximal half and treating distally located diametaphyseal fractures with K-wire fixation and proximally located fractures with ESIN defines a selection criterion for the method of stabilization. However, we were not able to confirm that this method proved helpful in reducing complications in our retrospective study. 

Novel, minimally invasive stabilization methods have been successfully applied in a limited number of patients, such as short double elastic nailing [38] or transepiphyseal placement of a single pre-bent K-wire (K-ESIN) [39]. During the study period, four children were treated with K-ESIN and two with transepiphyseal K-wire fixation without any complications. We refrain from drawing conclusions because of the limited number of patients treated with these novel stabilization methods. These new surgical stabilization methods might play a future role by supplementing the surgeon’s repertoire in cases where established methods prove problematic.

Our data suggest that treating a diametaphyseal fracture of the distal radius or forearm in children primarily involves CR and cast application, K-wire fixation, or ESIN. Surgeons should choose the stabilization method depending on the individual characteristics of the patient and fracture, family needs, and the surgeon’s preference.

The incidence of reoperations tended to be somewhat lower in the group of patients treated surgically than in the group of patients treated conservatively. We hypothesize that the trade-off for a less invasive procedure and shorter time of exposure to ionizing radiation is a slightly higher reoperation rate. However, it must be kept in mind that not every diametaphyseal fracture of the distal radius is suited to conservative treatment.

Stabilization of diametaphyseal fractures was conducted by many experienced pediatric surgical or pediatric orthopedic residents or consultants, which permits the generalizability of our results.

While our results might be relevant for busy fracture clinics treating children, our study has certain limitations. First, we used a retrospective study design, which may have resulted in several biases. Selection bias may have occurred because surgeons in charge selected fractures that appeared to be stable after CR for cast immobilization, with fractures considered unstable after CR preferably stabilized with K-wire fixation or ESIN. Given the retrospective nature of this study, we cannot exclude this confounder. Second, the number of included patients was moderate (n = 132), with some fixation methods contributing only a few patients (i.e., K-ESIN n = 4). Lastly, our patient data were collected at a single center. In order to achieve more robust results, future investigations should employ a prospective study design that incorporates randomization of patients to treatment methods and is carried out across multiple centers.

## 5. Conclusions

Our study indicated that the fixation method in treating diametaphyseal radius or forearm fractures in children is only of secondary concern. The overall success rate in retaining a stable position for the fragments in a diametaphyseal forearm fracture appears to be generally high. We hypothesize that fractures located at the diametaphyseal junction of the forearm in children can be treated by different methods of fixation, in accordance with the characteristics of the fracture, the demands of the patients and their families, and the surgeon’s preference. It should be kept in mind that K-wires can be removed in outpatient clinics without applying general anesthesia, whereas ESIN implants are usually removed in the operating room using general anesthesia in children.

The median total time of intraoperative exposure to image intensifier radiation did not differ significantly between surgical stabilization methods but was markedly lower in patients undergoing conservative treatment. Therefore, we opt for CR and primary cast application if circumstances allow, and parents are informed about the potential risks of treatment. Patients and their families must be made aware that the rate of secondary displacement tends to be higher after CR and cast immobilization than after surgical stabilization. The different options of surgical stabilization, i.e., K-wire fixation and ESIN, resulted in comparable rates of reoperations and complications.

Adequately powered, prospective multicenter studies are required to confirm our results. They may help decide if subdividing diametaphyseal fracture locations into proximal and distal fractures is of clinical relevance.

## Figures and Tables

**Figure 1 children-10-00374-f001:**
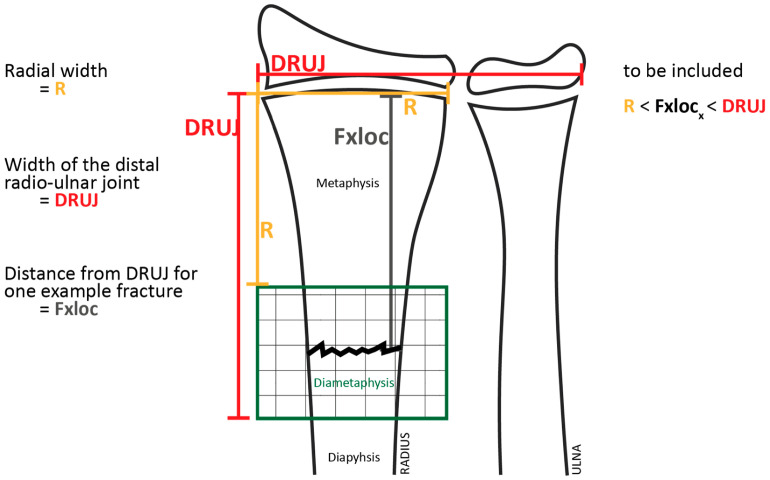
Definition of the diametaphyseal transition zone.

**Figure 2 children-10-00374-f002:**
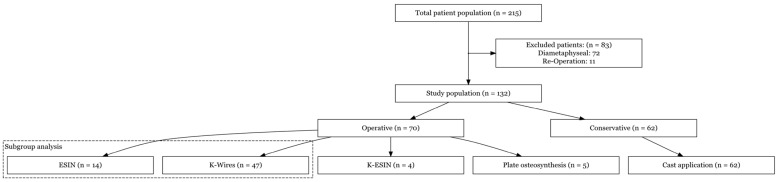
Flow chart of patient enrollment.

**Figure 3 children-10-00374-f003:**
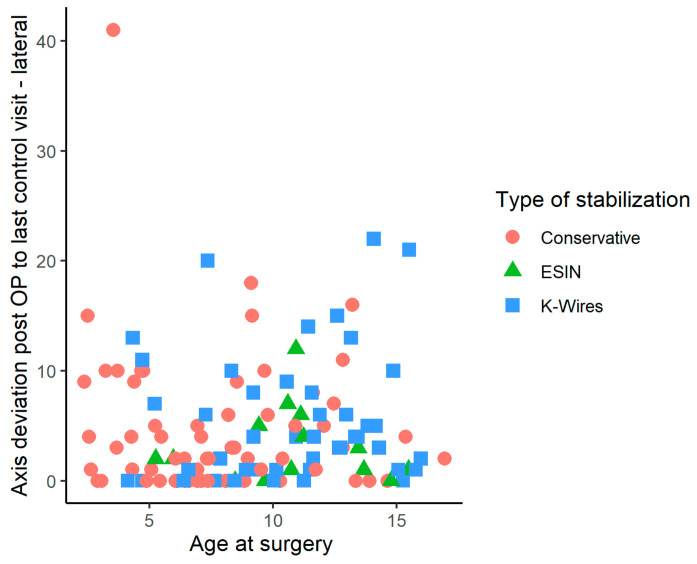
Change of axis deviation in lateral plane X-ray images obtained immediately after surgery, compared to axis deviations observed at the last follow-up, by age at surgery and the three types of stabilization.

**Figure 4 children-10-00374-f004:**
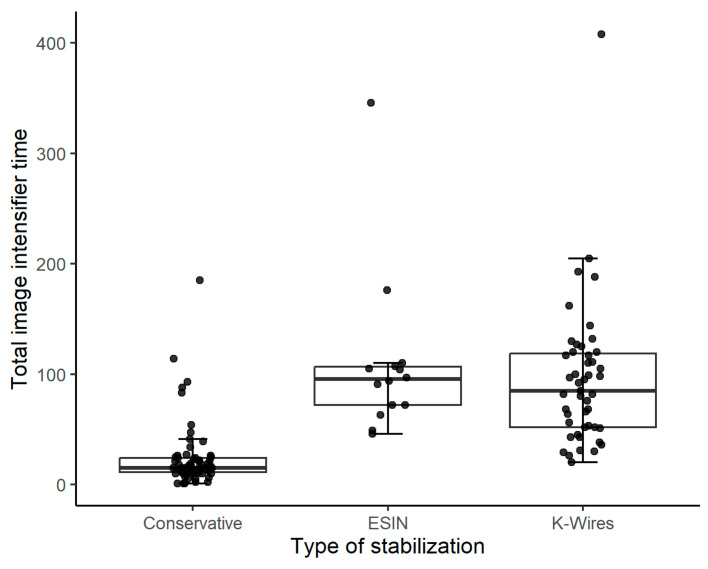
Total time of exposure to image intensifier radiation relative to the type of treatment (n = 127). Conservative treatment was applied in 62 patients, ESIN/K-ESIN in 18 (14/4) patients, and K-wire fixation in 47 patients.

**Figure 5 children-10-00374-f005:**
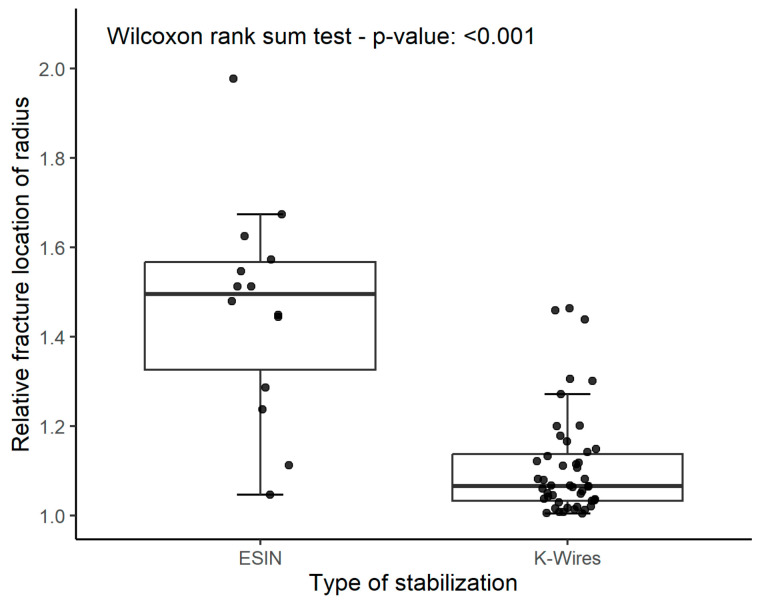
The boxplot graph shows that ESIN was significantly more often used in proximal fractures and K-wires more often in fractures located more distally within the diametaphysis of the distal radius.

**Figure 6 children-10-00374-f006:**
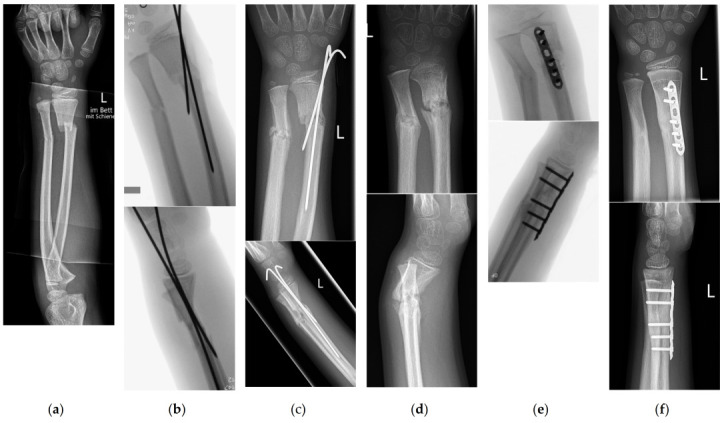
Boy, aged 9.5 years. He fell from a height of 1 m. (**a**) The image shows a completely displaced diametaphyseal fracture of the radius and a greenstick fracture of the distal shaft of the ulna. (**b**) Intraoperative image intensifier images during K-wire fixation of the diametaphyseal fracture of the radius. Note that a K-wire takes an intramedullary course within the proximal fragment, thereby reducing the stability of the fixation. (**c**) X-ray images obtained after 3 weeks show incomplete consolidation of fractures. K-wires were removed subsequently, and a forearm splint was applied. (**d**) X-ray images obtained after 7 weeks show a delayed fracture union and recurrent dorsal displacement of the radius fracture. (**e**) Open reduction and plate fixation were carried out 8 weeks after the injury. (**f**) Five months after the injury and 3 months after plate fixation, the fracture had healed.

**Table 1 children-10-00374-t001:** Patient data assessed.

Patients’ characteristics (gender, age); Mode of injury (fall in plain/height, sporting event, bicycle accident, or road traffic incident); Side of fracture (left, right, bilateral); Fracture location relative to maximum radial width and radio-carpal joint width; Fracture type (according to the AO/OTA Fracture and Dislocation Classification Compendium [26]); Type of reduction and stabilization (ESIN, K-ESIN, K-wires, open reduction, and internal fixation [ORIF] by volar plating, CR, and cast application); Number and characteristics of unplanned reoperations; Other complications (classified according to the Clavien-Dindo classification) [27,28]; Length of hospital stay (LOS); Radiologic axis deviation in lateral X-ray images (at initial presentation, from the first postoperative X-ray, and X-ray images obtained at the last follow-up); Angular stability (defined as the absolute magnitude of the angle difference between measurements of axial malalignment in lateral X-ray images obtained at the first postoperative visit and at the last follow-up); Total intraoperative image intensifier time; Radiologic data influencing fracture stability: ○ Both cortices of the radius are disrupted (on lateral images); ○ Displacement of more than one shaft width of the radius (in the lateral X-ray image); ○ Forearm fractures (combined fractures of radius and ulna).

**Table 2 children-10-00374-t002:** Fracture type by mode of injury.

	N Total	Spiral (%)	Oblique (%)	Transverse (%)	Multifragmentary/ Comminuted (%)
Fall	44	1 (2.3%)	18 (40.9%)	24 (54.5%)	1 (2.3%)
Sport	44	1 (2.3%)	16 (36.4%)	25 (56.8%)	2 (4.6%)
Fall from a height	24	3 (12.5%)	9 (37.5%)	12 (50%)	0
Bicycle accident	15	4 (26.7%)	3 (20%)	8 (53.3%	0
Traffic accident	3	0	0	2 (66.7%)	1 (33.3%)
Other/Unknown	2	0	1 (50%)	1 (50%)	0

**Table 3 children-10-00374-t003:** Treatment characteristics and unplanned reoperations (n = 132).

Characteristics	N (%)
Type of stabilization	
-Cast application	62 (47%)
-ESIN	14 (10.6%)
-K-wires	47 (35.6%)
-K-ESIN	4 (3.0%)
-Plate osteosynthesis	5 (3.8%)
Unplanned reoperation	
-Yes	11 (8.3%)
-No	121 (91.7%)

**Table 4 children-10-00374-t004:** Reoperations and complications by type of primary operations.

		N Total	Conservative (%) n = 62	Operative (%) n = 70	*p*-Value ^1^
Reoperation	Yes	11	7 (11.3%)	4 (5.7%)	0.347
(Clavien-Dindo IIIb)	No	121	55 (88.7%)	66 (94.3%)	
Other complications	Yes	19	6 (9.7%)	13 (18.6%)	0.214
	No	113	56 (90.3%)	57 (81.4%)	

^1^ Fisher’s exact test for count data.

**Table 5 children-10-00374-t005:** Complications graded according to Clavien-Dindo classification by type of stabilization [27,28].

Type of Stabilization	N Total	Clavien-Dindo (n)		*p*-Value ^1^
		Grade I	Grade IIIa	
Cast application	6	5	1	0.721
ESIN	3	2	1	
K-wires	10	9	1	

^1^ Fisher’s exact test for count data.

## Data Availability

All data are contained in the article. Additional data can be obtained from the corresponding author upon reasonable request.

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
