# Peer review of "Diametaphyseal Distal Forearm Fractures in Children: A STROBE Compliant Comparison of Outcomes of Different Stabilization Techniques Regarding Complications"

_children, 2023, doi:10.3390/children10020374_

Round 1

Reviewer 1 Report

The manuscript is a retrospective study aim to compare conservative and surgical stabilization of diametaphyseal forearm fractures in terms of their risks of unplanned secondary interventions requiring anesthesia. This retrospective analysis included 132 patients who had undergone treatment for diametaphyseal forearm fracture between 2013 and 2020 at our institution. The primary analysis compared complications occurring in patients treated conservatively with those in patients managed surgically (ESIN, K-wire fixation, KESIN stabilization, or open reduction and plate osteosynthesis). In a subgroup analysis, we compared the two most frequently applied surgical stabilization techniques in distal forearm fractures (i.e., ESIN and K-wire) with conservative treatment.

 I read the article with interest, the title is well thought out and faithfully reflects the content of the study. The abstract is adequately developed, and it is useful to frame the characteristics of the study.                                                                                                                                                                                       

In the introduction, foream fractures in children have been shortly described. In the materials and methods, have been adequately developed. The discussion is sufficiently developed.

Nevertheless, some minor changes are needed to be considered suitable for publication.

Comment 1: It would be more appropriate to add a short description of the purpose of the study.

Comment 2: In the introduction: It may be useful to include some bibliographical references more update on the great incidence of obesity in children. For example (Vescio A. et al (2022) "Is Obesity a Risk Factor for Loss of Reduction in Children with Distal Radius Fractures Treated Conservatively?") or (Pavone V. et al (2020) "Analysis of loss of reduction as risk factor for additional secondary displacement in children with displaced distal radius fractures treated conservatively")

Comment 3: In materials and methods: it would be appropriate to add some representative  X-ray images before and after fracture reduction and any redisplacements.

Comment 4: The limits of the study are clear enough. However, I would add a brief description of what could be improved in future studies on the same topic.

Comment 5: Finally, additional English editing is needed. The Non-Native Speakers of English Editing Certificate was not signed.

Author Response

Dear colleague,

thank you for taking the time to review our manuscript. Your comments and suggestions are greatly appreciated and were carefully considered in revising the paper.

In response to your remarks, I compiled the following list outlining our response to each comment (which is congruent with the attached document):

The manuscript is a retrospective study aim to compare conservative and surgical stabilization of diametaphyseal forearm fractures in terms of their risks of unplanned secondary interventions requiring anesthesia. This retrospective analysis included 132 patients who had undergone treatment for diametaphyseal forearm fracture between 2013 and 2020 at our institution. The primary analysis compared complications occurring in patients treated conservatively with those in patients managed surgically (ESIN, K-wire fixation, KESIN stabilization, or open reduction and plate osteosynthesis). In a subgroup analysis, we compared the two most frequently applied surgical stabilization techniques in distal forearm fractures (i.e., ESIN and K-wire) with conservative treatment.
I read the article with interest, the title is well thought out and faithfully reflects the content of the study. The abstract is adequately developed, and it is useful to frame the characteristics of the study.
In the introduction, foream fractures in children have been shortly described. In the materials and methods, have been adequately developed. The discussion is sufficiently developed.

We thank Reviewer 1 for these encouraging remarks.

Nevertheless, some minor changes are needed to be considered suitable for publication.

Comment 1: It would be more appropriate to add a short description of the purpose of the study.

Thank you for this helpful comment. We added a short description that helps comprehension.

Comment 2: In the introduction: It may be useful to include some bibliographical references more update on the great incidence of obesity in children. For example (Vescio A. et al (2022) "Is Obesity a Risk Factor for Loss of Reduction in Children with Distal Radius Fractures Treated Conservatively?") or (Pavone V. et al (2020) "Analysis of loss of reduction as risk factor for additional secondary displacement in children with displaced distal radius fractures treated conservatively")

We agree with Reviewer 1 that child obesity was underrepresented in the introduction and therefore included some background and the mentioned citations.

Comment 3: In materials and methods: it would be appropriate to add some representative X-ray images before and after fracture reduction and any redisplacements.

We found your comment very accurate and added some representative X-ray images to help the reader visualize the fracture reduction and its pitfalls.

Comment 4: The limits of the study are clear enough. However, I would add a brief description of what could be improved in future studies on the same topic.

We added suggestions on how to improve the study design.

Comment 5: Finally, additional English editing is needed. The Non-Native Speakers of English Editing Certificate was not signed.

The article's English language and style correction was revised by Silvia M. Rogers, PhD, of MediWrite, Basel, Switzerland, who had also proofread the original submitted manuscript.

Reviewer 2 Report

The subject is somewhat clear, and it has been explored much more than the current introduction gives credit. The article presents a good idea.  Although the initial question is interesting, I have a few issues with the study.

1. Title: good
2. Abstract: it captures the appropriate essence of the manuscript. Excellent.
3. Introduction: The introduction identifies the problem that is being addressed in the manuscript and develops and states the purpose of the manuscript.

4. Tables and figures: Quality of figures is so important too. Please provide some high-resolution figures. Some figures have a poor resolution.
5. References: I have verified all references and all key references are correct.

6. Methods: The Data used was appropriate.  

7. Discussion:

* The authors  discussed the  limitations of the study. I appreciate it
8. Conclusion: The conclusion is justified by the methods and results.
9. There are still some mistakes in grammar and misprints, the authors should carefully check this manuscript.

* I have enjoyed reading, and I am in favor of publication after suitable.

Author Response

Dear colleague,

thank you for taking the time to review our manuscript. Your comments and suggestions are greatly appreciated and were carefully considered in revising the paper.

In response to your remarks, I compiled the following list outlining our response to each comment (which is congruent with the attached document):

The subject is somewhat clear, and it has been explored much more than the current introduction gives credit. The article presents a good idea. Although the initial question is interesting, I have a few issues with the study.

  1. Title: good
  2. Abstract: it captures the appropriate essence of the manuscript. Excellent.
  3. Introduction: The introduction identifies the problem that is being addressed in the manuscript and develops and states the purpose of the manuscript.

Thank you for your encouraging remarks.

  1. Tables and figures: Quality of figures is so important too. Please provide some high-resolution figures. Some figures have a poor resolution.

We agree with Reviewer 2 that the resolution of the submitted version was subpar and replaced the figures with high-resolution .png-images. We will also include scalable vector graphics in our submission.

  1. References: I have verified all references and all key references are correct.

We thank Reviewer 2 for your thorough proofreading.

  1. Methods: The Data used was appropriate.
  2. Discussion: * The authors discussed the limitations of the study. I appreciate it
  3. Conclusion: The conclusion is justified by the methods and results.

We greatly appreciate your encouragement.

  1. There are still some mistakes in grammar and misprints, the authors should carefully check this manuscript.

Thank you for finding other mistakes which we eagerly corrected after another thorough proofreading by the study’s authors and Silvia M. Rogers, PhD, of MediWrite, Basel, Switzerland, who had also proofread the original submitted manuscript.

* I have enjoyed reading, and I am in favor of publication after suitable.
